

# Astrocyte activation in the anterior cingulate cortex and altered glutamatergic gene expression during paclitaxel-induced neuropathic pain in mice

Willias Masocha

Department Pharmacology and Therapeutics, Faculty of Pharmacy, Kuwait University, Safat, Kuwait

## ABSTRACT

Spinal astrocyte activation contributes to the pathogenesis of paclitaxel-induced neuropathic pain (PINP) in animal models. We examined glial fibrillary acidic protein (GFAP; an astrocyte marker) immunoreactivity and gene expression of GFAP, glutamate transporters and receptor subunits by real time PCR in the anterior cingulate cortex (ACC) at 7 days post first administration of paclitaxel, a time point when mice had developed thermal hyperalgesia. The ACC, an area in the brain involved in pain perception and modulation, was chosen because changes in this area might contribute to the pathophysiology of PINP. GFAP transcripts levels were elevated by more than fivefold and GFAP immunoreactivity increased in the ACC of paclitaxel-treated mice. The 6 glutamate transporters (GLAST, GLT-1 EAAC1, EAAT4, VGLUT-1 and VGLUT-2) quantified were not significantly altered by paclitaxel treatment. Of the 12 ionotropic glutamate receptor subunits transcripts analysed 6 (GLuA1, GLuA3, GLuK2, GLuK3, GLuK5 and GLuN1) were significantly up-regulated, whereas GLuA2, GLuK1, GLuK4, GLuN2A and GLuN2B were not significantly altered and GLuA4 was lowly expressed. Amongst the 8 metabotropic receptor subunits analysed only mGLuR8 was significantly elevated. In conclusion, during PINP there is astrocyte activation, with no change in glutamate transporter expression and differential up-regulation of glutamate receptor subunits in the ACC. Thus, targeting astrocyte activation and the glutamatergic system might be another therapeutic avenue for management of PINP.

## INTRODUCTION

The anterior cingulate cortex (ACC) is a cortical area in the brain that has been described to be involved with pain, possibly including both perception and modulation (*Vogt, 2005*; *Xie, Huo & Tang, 2009*; *Zhuo, 2008*). It is a component of the medial pain pathway. The afferent inputs to the ACC are from midline and intralaminar thalamic nuclei, whilst the ACC sends projections into various areas including the intralaminar thalamic nuclei and periaqueductal grey (PAG, which is involved in control of descending pain) (*Senapati*

Corresponding author
Willias Masocha,
masocha@hsc.edu.kw

*et al., 2005; Sewards & Sewards, 2002; Vogt, 2005*). Neuroimaging studies have shown increased activity in the ACC during chronic pain, including neuropathic pain (*Hsieh et al., 1995; Peyron, Laurent & Garcia-Larrea, 2000; Tseng et al., 2013*). Neurophysiological and molecular changes have also been observed in the ACC during chronic or neuropathic (*Wrigley et al., 2009; Xu et al., 2008; Yamashita et al., 2014*).

One of the changes that has been observed in the ACC during chronic or neuropathic pain is the activation of astrocytes or astrogliosis (*Chen et al., 2012; Kuzumaki et al., 2007; Lu, Zhu & Gao, 2011; Narita et al., 2006; Yamashita et al., 2014*). Astrocytes are the most numerous non-neuronal cells in the brain involved in modulation of neuronal activities e.g., extracellular and synaptic cleft neurotransmitter level regulation, release of neuroactive molecules amongst other activities (*Maragakis & Rothstein, 2006; Seifert, Schilling & Steinhauser, 2006*). Astrocytes express transporters which remove neurotransmitters such as $\gamma$-aminobutyric acid (GABA) and glutamate from the extracellular space or synaptic cleft (*Conti et al., 1998; Danbolt, 2001; Gosselin, Bebber & Decosterd, 2010; Minelli et al., 1995; Wang & Bordey, 2008*). Astrocyte activation has been linked with increase in transporters for GABA and a decrease in transporters for glutamate resulting in a more excitatory state in the brain (*Gosselin, Bebber & Decosterd, 2010; Maragakis & Rothstein, 2006*). Recently, we observed an increase in the transcripts of GABA transporter 1 (GAT-1) in a rodent model of paclitaxel-induced neuropathic pain (PINP) (*Masocha, 2015*). However, it is not known whether paclitaxel induces astrocyte activation in the ACC although it has been shown to induce astrocyte activation in the spinal cord (*Peters et al., 2007; Zhang, Yoon & Dougherty, 2012*). Paclitaxel is a chemotherapeutic agent that causes dose-dependent neuropathic pain in some patients (*Scripture, Figg & Sparreboom, 2006; Wolf et al., 2008*). In the rodent models, we observed that the PINP is linked with disturbances in the GABAergic system (*Masocha, 2015*) resulting in increased excitability of the ACC to electrophysiological stimulation (H Nashawi, IO Edafiogho, SB Kombian, W Masocha, 2015, unpublished data). GABA is the major inhibitory neurotransmitter while glutamate is the major stimulatory neurotransmitter in the brain (*Meldrum, 2000; Petroff, 2002*). It is not known whether paclitaxel causes any changes in the glutamatergic system in the ACC, although it has been shown to decrease the expression of glutamate transporters such as GLAST and GLT-1 in the spinal cord (*Weng et al., 2005; Zhang, Yoon & Dougherty, 2012*). There are 8 known glutamate transporters, which are excitatory amino acid transporter 1 (EAAT1; referred to as GLAST in rodents), EAAT2 (GLT-1), EAAT3 (EAAC1), EAAT4, EAAT5, vesicular glutamate transporter 1 (VGLUT1), VGLUT2, and VGLUT3 (*Danbolt, 2001; Shigeri, Seal & Shimamoto, 2004*). Of the transporters, GLAST and GLT-1 are expressed on astrocytes (*Danbolt, 2001*) and play an important role in removal of glutamate from the synaptic cleft and extracellular space (*Danbolt, 2001; Shigeri, Seal & Shimamoto, 2004*); if their expression is down-regulated, this results in increased levels of glutamate and excitotoxicity (*Danbolt, 2001; Rothstein et al., 1996; Shigeri, Seal & Shimamoto, 2004; Yi, Pow & Hazell, 2005*). Glutamate acts on ionotropic and metabotropic receptors. The ionotropic receptors are divided into alpha-amino-3-hydroxy-5-methyl-4-isoxazolpropionate (AMPA), kainate and N-methyl-D-aspartate

(NMDA) receptors which have 18 subunits GLuA1 to 4, GLuK1-5 and GLuN1, GLuN2A to D, GLuN3A and B, and GLuD1 and 2 (*Collingridge et al., 2009*). There are 8 subunits of the metabotropic receptors mGLUR$_1$ to $_8$ (*Conn & Pin, 1997*; *Niswender & Conn, 2010*).

Astrocyte activation, which has been observed in the ACC in models of chronic and neuropathic pain (*Chen et al., 2012*; *Kuzumaki et al., 2007*; *Lu, Zhu & Gao, 2011*; *Narita et al., 2006*; *Yamashita et al., 2014*), might occur in the ACC during PINP together with molecular changes in the glutamatergic system contributing to the pathogenesis or maintenance of PINP. Thus, in this study, astrocyte activation and the gene expression of molecules of the astrocyte marker (glial fibrillary acidic protein (GFAP), glutamate transporters and receptors in the ACC were evaluated in mice at a time point when the mice had paclitaxel-induced thermal hyperalgesia (*Nieto et al., 2008*; *Parvathy & Masocha, 2013*).

# MATERIALS AND METHODS

## Animals
Ninety eight female BALB/c mice (8–12 weeks old) supplied by the Animal Resources Centre (ARC) at the Health Sciences Center (HSC), Kuwait University were used. The animals were housed and handled in compliance with the Kuwait University, HSC, ARC guidelines and published ethical guidelines for research in experimental pain with conscious animals (*Zimmermann, 1983*). All animal experiments were approved by the Ethical Committee for the use of Laboratory Animals in Teaching and in Research, HSC, Kuwait University.

## Paclitaxel administration
Paclitaxel (Cat. No. 1097; Tocris, Bristol, UK) was dissolved in a solution made up of 50% Cremophor EL and 50% absolute ethanol to a concentration of 6 mg/ml and then diluted in normal saline (NaCl 0.9%), to a final concentration of 0.2 mg/ml just before administration. The vehicle for paclitaxel, thus, constituted of about 1.7% Cremophor EL and 1.7% ethanol in normal saline. Paclitaxel 2 mg/kg or its vehicle were administered to mice intraperitoneally (i.p.), daily for 5 consecutive days. This treatment regimen has been reported to produce painful neuropathy and thermal hyperalgesia in mice (*Nieto et al., 2008*; *Parvathy & Masocha, 2013*).

## Hot plate test
Reaction latencies to hot plate test were measured before (baseline latency) and on day 7 after first administration of paclitaxel. Briefly, mice were placed on a hot plate (Panlab SL, Barcelona, Spain) with the temperature adjusted to $55 \pm 1\,°C$. The time to the first sign of nociception, paw licking or flinching, was recorded and the animal immediately removed from the hot plate. A cut-off period of 20 s was maintained to avoid damage to the paws.

## ACC tissue preparation
The mice were anesthetized with isoflurane and sacrificed by decapitation. ACC was dissected and prepared for RNA extraction on day 7 post-first administration of paclitaxel—a time point when mice had developed thermal hyperalgesia (*Parvathy & Masocha, 2013*)—as previously described (*Masocha, 2015*).

**Table 1  PCR primer sequences of cyclophilin, GFAP and glutamatergic system molecules.**

| Gene | Polarity | |
| --- | --- | --- |
| | Sense<br>Sequence 5′–3′ | Anti-sense<br>Sequence 5′–3′ |
| Cyclophilin | GCTTTTCGCCGCTTGCT | CTCGTCATCGGCCGTGAT |
| GFAP | ACAGCGGCCCTGAGAGAGAT | CTCCTCTGTCTCTTGCATGTTACTG |
| GLAST | ACCAAAAGCAACGGAGAAGAG | GGCATTCCGAAACAGGTAACTC |
| GLT-1 | ACAATATGCCCAAGCAGGTAGA | CTTTGGCTCATCGGAGCTGA |
| EAAC1 | CTTCCTACGGAATCACTGGCT | CGATCAGCGGCAAAATGACC |
| EAAT4 | AGCAGCCACGGCAATAGTC | ATGCCAAGCTGACACCAATGA |
| VGLUT-1 | GGTGGAGGGGGTCACATAC | AGATCCCGAAGCTGCCATAGA |
| VGLUT-2 | CCCTGGAGGTGCCTGAGAA | GCGGTGGATAGTGCTGTTGTT |
| GLuA1 | CCGTTGACACATCCAATCAGTTT | GTCGATAATGCTAATGAGAGCTTCCT |
| GLuA2 | AAATTGCCAAACATTGTGG | ATGGAGCCATGGCAATATCA |
| GLuA3 | ACACCATCAGCATAGGTGGA | TCAGTGGTGTTCTGGTTGGT |
| GLuA4 | TTGGAATGGGATGGTAGGAG | TAGGAACAAGACCACGCTGA |
| GLuK1 | TCACACCCTACGAGTGGTATAAC | AGCTCCAACGCCAAACCAG |
| GLuK2 | ATCGGATATTCGCAAGGAACC | CCATAGGGCCAGATTCCACA |
| GLuK3 | AGGTCCTAATGTCACTGACTCTC | GCCATAAAGGGTCCTATCAGAC |
| GLuK4 | CCAAGGTCGAAGTGGACATCT | CTGGGGGTGAAGGTTCAGGG |
| GLuK5 | ATAGTCGCCTTCGCCAATCC | GTGTCCGTGGTCTCGTACTG |
| GLuN1 | GGCATCGTAGCTGGGATCTTC | TCCTACGGGCATCCTTGTG |
| GLuN2A | GTTTGTTGGTGACGGTGAGA | AAGAGGTGCTCCCAGATGAA |
| GLuN2B | ATGTGGATTGGGAGGATAGG | TCGGGCTTTGAGGATACTTG |
| mGluR1 | TGTCATCAACGCCATCTATGC | CCCACGTAGCCAGGACATAGAG |
| mGluR2 | CGCTCTCTGCACGCTCTATG | GATGAACTTGGCCTCGTTGAA |
| mGluR3 | AAGCCATCGCCTGTCATCTG | GGAGGTCCCAAGCCCAAGT |
| mGluR4 | GATGCTCTACATGCCCAAAGTCTAC | CGGTGACAACGGCTTTGAG |
| mGluR5 | TGACCCTGAGCCCATTGC | AACGAAGAGGGTGGCTAGCA |
| mGluR6 | TCATGGCCACCACAACTATCA | CAGAGGCGCGGACTATGG |
| mGluR7 | AAGCCTGGGCAGAGGAAGA | TCCATCACAGGGCTCACAAG |
| mGluR8 | CAGCATCTGTCTGCAGCCTG | CGGTTTTCTTCCTCTCCCCA |

## Real time RT-PCR

Gene transcripts of the astrocyte marker GFAP, 6 glutamate transporters (GLAST, GLT-1, EAAC1, EAAT4, VGLUT1, VGLUT2), 12 ionotropic glutamate receptor subunits (GLuA1 to 4, GLuK1 to 5, GLuN1, GLuN2A and GLuN2B) and 8 metabotropic glutamate subunits (mGluR$_1$ to $_8$) were quantified in the ACC of vehicle-treated or paclitaxel-treated by real time PCR. Total RNA was extracted from the fresh frozen ACC using the RNeasy Kit (Qiagen GmbH, Venlo, Netherlands), reverse-transcribed, and the mRNA levels were quantified on an ABI Prism® 7500 sequence detection system (Applied Biosystems, Carlsbad, California, USA) as previously described (*Masocha, 2009*). The primer sequences which were used, listed in Table 1, were ordered from Invitrogen (Life Technologies, Carlsbad, California, USA) and/or synthesized at the Research Core Facility (RCF), HSC, Kuwait University. The amplification and detection were performed as follows: a first hold

at 50 °C for 2 min, a second hold at 95 °C for 2 min followed by 40 cycles at 95 °C for 15 s and 63 °C for 1 min. Threshold cycle (Ct) values for all cDNA samples were obtained and the amount of mRNA of individual animal sample ($n = 6$–24 per group) was normalized to cyclophilin (housekeeping gene) ($\Delta$Ct). The relative amount of target gene transcripts was calculated using the $2^{-\Delta\Delta Ct}$ method as described previously (*Livak & Schmittgen, 2001*).

## Immunohistochemistry

Fresh-frozen brains were cut on a cryostat into 25 μm thick sections and thaw-mounted on chrome-alum gelatin–coated slides. The sections at a level of the lateral ventricles and the ACC were fixed in 4% formalin and 14% picric acid in PBS for 30 s at 4 °C, rinsed in PBS, fixed in acetone for 30 s at −20 °C, and then rinsed in PBS. All sections were preincubated with 1% bovine serum albumin and 0.3% Triton X-100 in PBS (solution used as diluent for primary and secondary antibodies) for 30 min at room temperature. Sections were incubated with rabbit anti- GFAP (1:100; DAKO, Glostrup, Denmark) for 2 h at room temperature to immunostain astrocytes. Sections were then rinsed in PBS and incubated with DyLight 594-conjugated Affinipure donkey Anti-rabbit IgG (H + L) (1:100, Jackson ImmunoResearch Laboratories, West Grove, Pennsylvania, USA) for 1 h. The sections were rinsed in PBS and mounted in ProLong® Gold antifade reagent (Invitrogen, USA). Sections were examined and analysed using a LSM 700 laser scanning confocal microscope. Images were taken from the ACC using an Axio imager (Carl Zeiss MicroImaging GmbH, Oberkochen, Germany).

## Statistical analyses

Statistical analyses were performed using unpaired two-tailed Student's *t*-test using Graph Pad Prism software (version 5.0). The differences were considered significant at $p < 0.05$. The results in the text and figures are expressed as the means ± S.E.M.

# RESULTS

## Paclitaxel-induced thermal hyperalgesia

Mice developed thermal hyperalgesia on day 7 after first administration of paclitaxel as we previously described (*Masocha, 2014*; *Parvathy & Masocha, 2013*) i.e., paclitaxel-treated mice had significant reduction in response latency time in the hot plate test on day 7 compared to the baseline latency and vehicle-treated animals (6.23 ± 0.28 s compared to 9.66 ± 0.16 s and 9.00 ± 0.38 s, respectively; $n = 10$ vehicle-treated mice and 16 paclitaxel treated-mice; $p < 0.05$ for both comparisons).

## Astrocyte activation in the ACC at 7 days after paclitaxel administration

The mRNA expression and immunoreactivity of the astrocyte marker, GFAP, were analysed in the ACC at day 7, a time when the mice had developed thermal hyperalgesia. Treatment with paclitaxel significantly increased the expression of GFAP transcripts ($p = 0.02$) by more than fivefold compared to vehicle-treated controls (Fig. 1). Confocal microscopy images showed that in paclitaxel-treated mice there was increased GFAP immunoreactivity

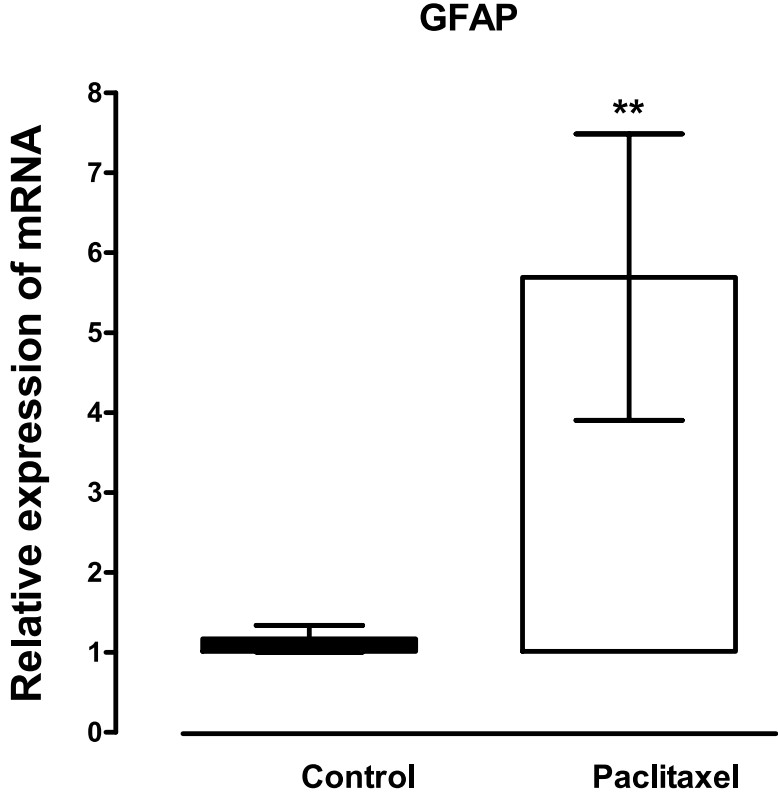

**Figure 1** **Effects of paclitaxel on glial fibrillary acidic protein (GFAP) transcript levels in the anterior cingulate cortex (ACC).** Relative GFAP mRNA expression in the ACC of BALB/c mice on day 7 after first administration of the drug or its vehicle. Each point represents the mean ± S.E.M of the values obtained from 21 vehicle-treated control mice and 24 paclitaxel-treated mice. ** $p < 0.01$ compared to vehicle-treated control mice.

in the ACC compared to vehicle-treated controls (Fig. 2). However, the change in GFAP immunoreactivity in paclitaxel-treated animals varied across the ACC and animals i.e., it was not robust in all animals and did not cover most of the ACC.

## Expression of transcripts of glutamate transporters in the ACC at 7 days after paclitaxel administration

There were no differences observed in the transcript levels of all the six glutamate transporters analysed (Fig. 3) in the ACC of paclitaxel-treated mice compared to vehicle-treated mice. Using the unpaired two-tailed Student's $t$-test the $p$ values obtained are: 0.7243 for GLAST, 0.6608 for GLT-1, 0.7575 for EAAC1, 0.5925 for EAAT4, 0.8885 for VGLUT-1 and 0.0858 for VGLUT-2.

## Expression of transcripts of glutamate receptors in the ACC at 7 days after paclitaxel administration

Amongst the AMPA receptor subunits GLuA4 was lowly expressed in the ACC and mRNA expression was not detected after 40 cycles in the real time RT-PCR in 12 out of 16 vehicle- and paclitaxel-treated animals analysed. Treatment with paclitaxel did not significantly alter the mRNA expression of the AMPA receptor subunit GLuA2 ($p = 0.9720$), but

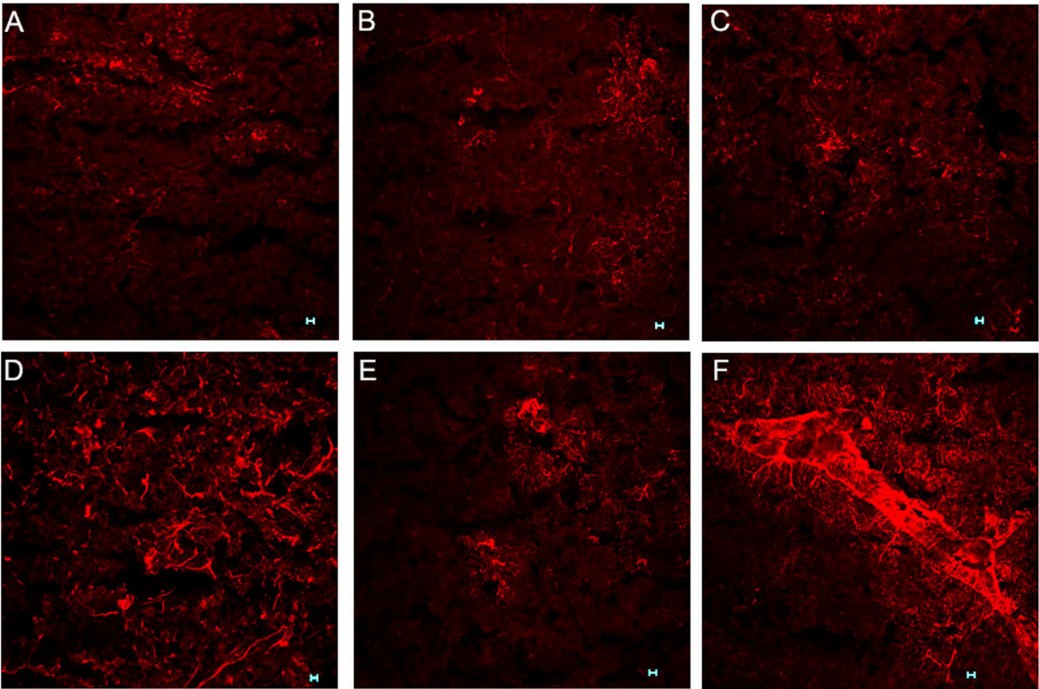

**Figure 2 Effects of paclitaxel on glial fibrillary acidic protein (GFAP) immunoreactivity in the anterior cingulate cortex (ACC).** GFAP immunoreactivity in the ACC of BALB/c mice on day 7 after first administration of the drug or its vehicle. GFAP immunoreactivity in astrocytes is increased in 3 paclitaxel-treated mice (D–F) compared to 3 vehicle-treated control mice (A–C) in the ACC. Note that in a paclitaxel-treated mouse (D) increased immunoreactivity of GFAP appears to be along a blood vessel: Scale bar: 50 μm.

significantly increased the expression of GLuA1 ($p = 0.0166$) and GLuA3 ($p = 0.0243$) subunits compared to vehicle-treated controls (Fig. 4A).

Amongst the 5 kainate receptor subunits analysed, treatment with paclitaxel significantly increased the expression of the 3 subunits GluK2 ($p = 0.0136$), GluK3 ($p = 0.0026$) and GluK5 ($p = 0.0011$), but not 2 subunits GluK1 ($p = 0.4367$) and GluK4 ($p = 0.2785$), compared to vehicle-treated controls (Fig. 4B).

Amongst the 3 NMDA receptor subunits analysed, treatment with paclitaxel significantly increased the expression of GluN1 ($p = 0.0209$) only, but not 2 subunits GluN2A ($p = 0.0612$) and GluN2B ($p = 0.1105$), compared to vehicle-treated controls (Fig. 4C).

Of all the eight metabotropic glutamate receptors subunits quantified, only mGLuR$_8$ was significantly altered ($p = 0.0144$) in the ACC by treatment with paclitaxel compared to treatment with vehicle (Figs. 4E and 4F). Using the unpaired two-tailed Student's $t$-test, the $p$ values obtained are: 0.4439 for mGLuR$_1$, 0.1340 for mGLuR$_2$, 0.3201 for mGLuR$_3$, 0.9971 for mGLuR$_4$, 0.3375 for mGLuR$_5$, 0.9693 for mGLuR$_6$ and 0.2780 for mGLuR$_7$.

## DISCUSSION

This is the first study to report on the quantification and/or changes in the transcript levels and immunoreactivity of the astrocyte marker GFAP, transcript levels of glutamate transporters and receptors in the ACC, an area associated with pain perception and

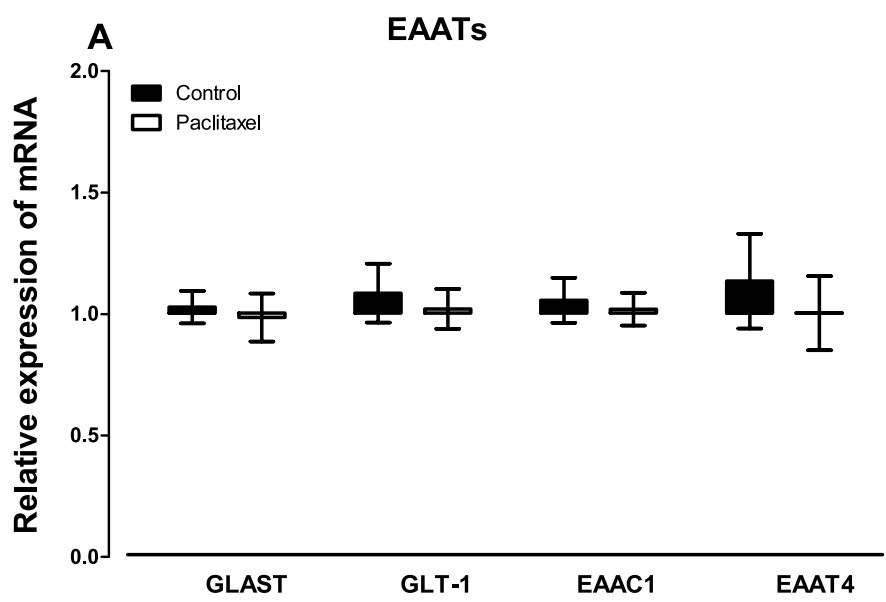

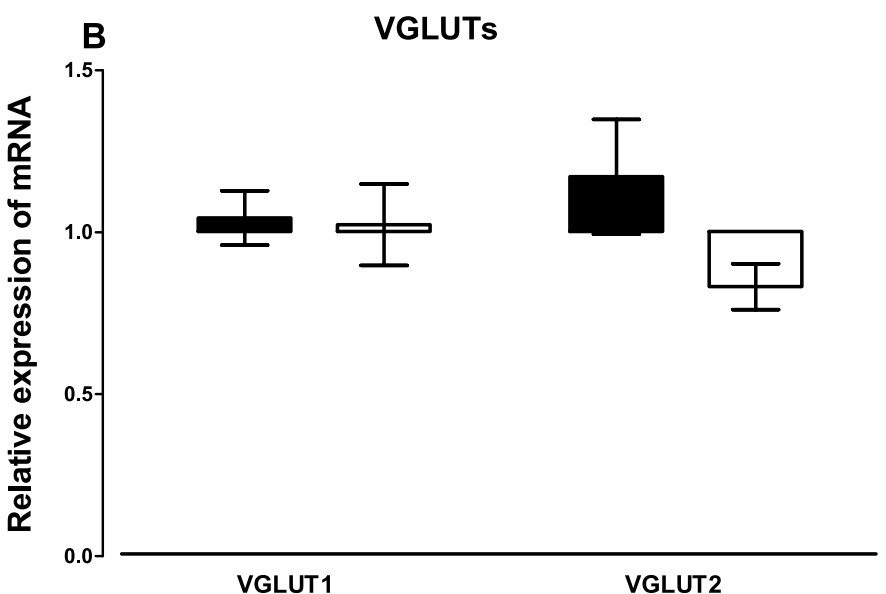

**Figure 3 Effects of paclitaxel on glutamate transporters transcript levels in the anterior cingulate cortex (ACC).** Relative mRNA expression of (A) excitatory amino acid transporters GLAST, GLT-1, EAAC1, EAAT4, and (B) vesicular glutamate transporters VGLUT1 and VGLUT2 in the ACC of BALB/c mice on day 7 after first administration of the drug or its vehicle. Each point represents the mean ± S.E.M of the values obtained from 11 to 15 vehicle-treated control mice and 13–15 paclitaxel-treated mice.

modulation (*Vogt, 2005*; *Xie, Huo & Tang, 2009*; *Zhuo, 2008*), during paclitaxel-induced neuropathic pain (PINP).

Increased expression of GFAP in the brain is a marker of astrocyte activation (*Aldskogius & Kozlova, 1998*). Various studies have reported increased expression of GFAP mRNA and protein in the ACC during pain (*Chen et al., 2012*; *Kuzumaki et al., 2007*; *Lu, Zhu &*

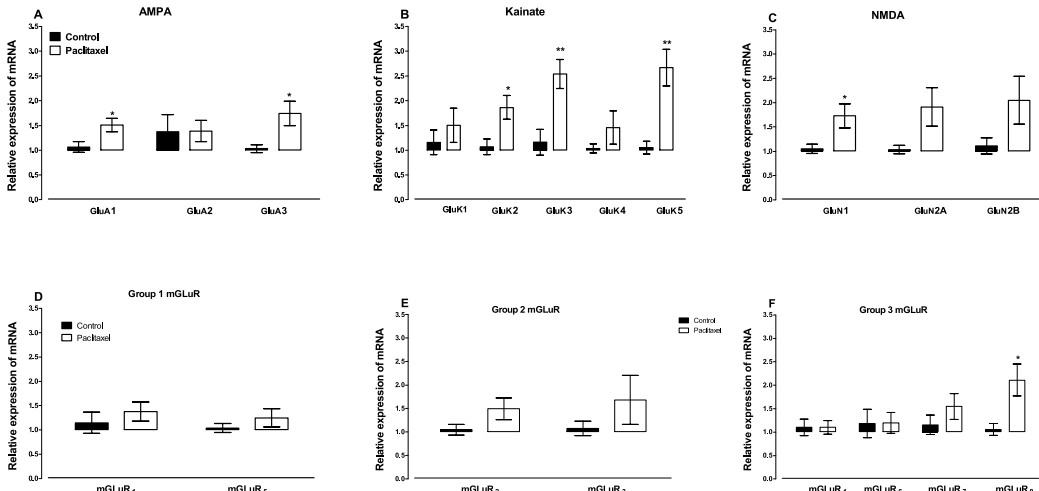

**Figure 4  Effects of paclitaxel on glutamate receptors transcript levels in the anterior cingulate cortex (ACC).** Relative mRNA expression of (A) AMPA receptor subunits GLuA1 to 3, (B) kainate receptor subunits GLuK1 to 5, (C) NMDA receptor subunits GLuN1, GLuN2A and GLuN2B, and (D–F) metabotropic glutamate receptors mGLuR$_1$ to $_8$ in the ACC of BALB/c mice on day 7 after first administration of the drug or its vehicle. Each point represents the mean $\pm$ S.E.M of the values obtained from 6 to 15 vehicle-treated control mice and 8–16 paclitaxel-treated mice. *$p < 0.05$, **$p < 0.01$ compared to vehicle-treated control mice.

*Gao, 2011*). Astrocyte activation has also been observed in the ACC in other models of neuropathic pain (*Xu et al., 2008*; *Yamashita et al., 2014*) but had not been reported in PINP. However, astrocyte activation in the spinal cord has been reported to contribute to PINP in rodents (*Ruiz-Medina et al., 2013*; *Zhang, Yoon & Dougherty, 2012*). In the current study, the expression of GFAP transcripts and immunoreactivity in the ACC was increased in mice treated with PINP. During peripheral nerve injury neurons have been reported to release neurotransmitters such substance *P* and glutamate and neuronal chemokines that cause astrocyte activation in the CNS (*Milligan & Watkins, 2009*; *Wang et al., 2009*; *Watkins et al., 2007*). Activated astrocytes in turn release molecules that contribute to the pathophysiology of pain through modulation of neuronal functioning (*Milligan & Watkins, 2009*; *Wang et al., 2009*; *Watkins et al., 2007*). Thus, the current results suggest that astrocyte activation in the ACC might also contribute to the pathophysiology of PINP.

The activation of astrocytes in the spinal cord induced by paclitaxel has been reported to be accompanied with a decrease in the expression of the glial glutamate transporters GLAST and GLT-1 (*Zhang, Yoon & Dougherty, 2012*) as well as an increase in the GABA transporter GAT-1 (*Yadav et al., 2015*). In the current study, there were no changes in the transcript levels of glutamate transporters in the ACC of paclitaxel-treated mice. However, in a recent study, we observed elevated transcripts of GAT-1 in the ACC of mice with PINP (*Masocha, 2015*). This suggests that astrocyte activation and increased expression of GAT-1, but not glutamate transporters, in the ACC play a role in the pathogenesis in PINP. This would result in an imbalance in the inhibitory (GABA) and excitatory (glutamate) neurotransmitters, which might result in increased excitability of the ACC. Increased neuronal excitability in the ACC might contribute to the increased activity observed in

the ACC during neuropathic pain in both humans and animal models (*Hsieh et al., 1995*; *Peyron, Laurent & Garcia-Larrea, 2000*; *Tseng et al., 2013*; *Wrigley et al., 2009*; *Xu et al., 2008*; *Yamashita et al., 2014*).

Although we did not observe any changes in the glutamate transporters in the ACC, we observed that transcripts of various glutamate receptors and receptor subunits were elevated in the ACC of mice with PINP. The increased expression of some of the glutamate receptors and receptor subunits could have been linked to astrocyte activation since all of the up-regulated receptors are expressed on astrocytes (*Geurts et al., 2005*; *Martínez-Lozada & Ortega, 2015*). Several receptors have been reported to be differentially expressed in the ACC in rodent models of PINP. We observed an increase in the expression of various GABA receptors in the ACC during PINP (*Masocha, 2015*). *Ortega-Legaspi et al. (2011)* and *Ortega-Legaspi et al. (2010)*. reported a differential expression of muscarinic-1 and −2 receptors and dopamine D1 and D2 receptors in the ACC of rodents with PINP. The increased expression glutamate receptors in the ACC also suggest a role of the glutamatergic system in the pathogenesis of PINP.

## CONCLUSIONS

In conclusion, the results of this study show that animals with paclitaxel-induced neuropathic pain (PINP) have increased transcripts and immunoreactivity of the astrocyte marker GFAP and transcripts of some glutamate receptors and receptor subunits, but not glutamate transporters, in the ACC. In a previous study, transcripts of a GABA transporter GAT-1, whose increase has been associated with astrocyte activation in the spinal cord of rodents with PINP (*Yadav et al., 2015*), was found increased in the ACC of mice with PINP (*Masocha, 2015*). Thus, inhibition of astrocyte activation and GAT-1 activity and/or antagonism of specific glutamate receptors could be therapeutic modalities of managing PINP and possibly other types on chemotherapy-induced neuropathic pain.

## ACKNOWLEDGEMENTS

I am grateful to Dr. Subramanian S Parvathy, Ms. Salini Soman from the Department of Pharmacology and Therapeutics, Faculty of Pharmacy, and Ms. Jucy Gabriel from the Research Core Facility, HSC, Kuwait University for their technical assistance and to the staff from the Animal Resources Centre, HSC, Kuwait University for their support.

### Funding

This study was supported by grants PT01/09 and SRUL02/13 from Kuwait University Research Sector. The funders had no role in study design, data collection and analysis, decision to publish, or preparation of the manuscript.

### Grant Disclosures

The following grant information was disclosed by the author:
Kuwait University Research Sector: PT01/09, SRUL02/13.

## Competing Interests

The authors declare there are no competing interests.

## Author Contributions

- Willias Masocha conceived and designed the experiments, performed the experiments, analyzed the data, contributed reagents/materials/analysis tools, wrote the paper, prepared figures and/or tables, reviewed drafts of the paper.

## Animal Ethics

The following information was supplied relating to ethical approvals (i.e., approving body and any reference numbers):

All animal experiments were approved by the Ethical Committee for the use of Laboratory Animals in Teaching and in Research, HSC, Kuwait University.

## Supplemental Information

Supplemental information for this article can be found online at http://dx.doi.org/10.7717/peerj.1350#supplemental-information.

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
