# Peer review of "Astrocyte activation in the anterior cingulate cortex and altered glutamatergic gene expression during paclitaxel-induced neuropathic pain in mice"

_PeerJ, doi:10.7717/peerj.1350_

## Round 0.1 · original submission · Major Revisions

Dear Authors, There are many comments that must be resolved before the manuscript can be resubmitted especially those from Reviewer 1 and 2.

Reviewer 1 ·

Basic reporting

.

Experimental design

.

Validity of the findings

The results provided are not enough to support the conclusion. The following points must be addressed.

1. Whether thermal hyperalgesia was developed in the experiment should be shown in the results.
2. One of the major results of this study is upregulation of GFAP in the anterior cingulate cortex. A immunocytochemical staining showing this result is necessary.

Additional comments

There were some few grammar and typing mistakes.

·

Basic reporting

No Comments

Experimental design

No Comments

Validity of the findings

No comments

Additional comments

Author attempted to extend his work on neuropathic pain induced by paclitaxel and evaluate alteration in glutamatergic gene expression ACC of mice. This is first study to examine paclitaxel effect on glutamatergic gene expression in ACC. This study would be beneficial role in neuropathic pain and therapeutic approach in future. This study is well suitable for publication in PeerJ. However, author has to modify some changes before consider for publication.
After abstract, key word did not include. Introduction was written well, but there is no literature about ACC afferent and efferent innervations in nociceptive pathway. Hypothesis did not write well.
P6, L 74 split subheadings in respective materials and methods. Animal groups and number of animals in each group did not classified well, animals ethical clearance number should be included.
Why author used specifically female mice for this study? Why not male?. Paclitaxel catalog number and how much total mg of paclitael dissolved? After behaviour experiment, How animals were sacrificed?
Although, author published Paclitaxel developed thermal hyperalgesia at day 7 in previous studies, in the present work, behavior method should be included. Also, In result column, it would be better to include behavior data of thermal hyperalgesia induced by paclitaxel.
Results and discussion should write separately. Discussion is not written well. Author need to discuss with other neurotransmitters role on astrocytes expression in pain pathogenesis.
Did author find any morphological changes in ACC after administration of Paclitaxel. In legends, number of animals in control and treated should be mentioned.
Along with primer sequence better to mentioned how many thermal cycles and temperature used for in each and every genes in RT-PCR. Why author did not show gel documentation at least some results.
This whole study depends upon the gene expression. Why did not author use housekeeping gene (Glyceraldehyde 3-phosphate dehydrogenase (GAPDH ) for positive control to compare with control treated one.

---

## Round 0.2 · accepted · Accept

Dear Author,

Your revised manuscript has been accepted by both peer reviewers.Congratulations! The manuscript will undergo further galley proof processing.

Reviewer 1 ·

Basic reporting

no further comments

Experimental design

no further comments

Validity of the findings

no further comments

Additional comments

N/A

·

Basic reporting

No Comments

Experimental design

No Comments

Validity of the findings

No Comments

Additional comments

No Comments